# E3 Ubiquitin Ligase CHIP Inhibits Haemocyte Proliferation and Differentiation via the Ubiquitination of Runx in the Pacific Oyster

**DOI:** 10.3390/cells13181535

**Published:** 2024-09-13

**Authors:** Miren Dong, Ying Song, Weilin Wang, Xiaorui Song, Wei Wu, Lingling Wang, Linsheng Song

**Affiliations:** 1Liaoning Key Laboratory of Marine Animal Immunology, Dalian Ocean University, Dalian 116023, China; dongmiren723@163.com (M.D.); song15ying@163.com (Y.S.); wangweilin@dlou.edu.cn (W.W.); songxiaorui@dlou.edu.cn (X.S.); wuwei12292024@163.com (W.W.); 2Liaoning Key Laboratory of Marine Animal Immunology and Disease Control, Dalian Ocean University, Dalian 116023, China; 3Dalian Key Laboratory of Aquatic Animal Disease Prevention and Control, Dalian Ocean University, Dalian 116023, China; 4Functional Laboratory of Marine Fisheries Science and Food Production Process, Qingdao National Laboratory for Marine Science and Technology, Qingdao 266235, China

**Keywords:** *Crassostrea gigas*, haemocytes, CHIP, Runx, differentiation, proliferation

## Abstract

Mollusca first evolve primitive immune cells (namely, haemocytes), which assemble a notable complex innate immune system, which are continuously produced through proliferation and differentiation and infused in the haemolymph. As a typical E3 ligase, CHIP is critical for immune cell turnover and homeostasis in vertebrates. In this study, a CHIP homolog (*Cg*CHIP) with a high expression in haemocytes was identified in oysters to investigate its role in the proliferation and differentiation of ancient innate immune cells. *Cg*CHIP exhibited a widespread distribution across all haemocyte subpopulations, and the knockdown of *Cg*CHIP altered the composition of haemocytes as examined by flow cytometry. Mechanistically screened with bioinformatics and immunoprecipitation, a key haematopoietic transcription factor *Cg*Runx was identified as a substrate of *Cg*CHIP. Moreover, amino acids in the interacted intervals of *Cg*CHIP and *Cg*Runx were determined by molecular docking. Experimental evidence from an in vitro culture model of an agranulocyte subpopulation and an in vivo oyster model revealed that the knockdown of *Cg*CHIP and *Cg*Runx had opposing effects on agranulocyte (precursor cells) differentiation and granulocyte (effector cells) proliferation. In summary, *Cg*CHIP negatively regulated agranulocyte differentiation and granulocyte proliferation by mediating the ubiquitination and degradation of *Cg*Runx in oysters. These results offer insight into the involvement of ubiquitylation in controlling haemocyte turnover in primitive invertebrates.

## 1. Introduction

Immune cells undergo continuous cellular turnover driven by proliferation and differentiation to replace old or damaged cells for maintaining immune homeostasis [1]. These processes are essential for replenishing all immune cell lineages and establishing the immune defense system, and thus, are tightly regulated [2,3]. Emerging reports provide insights into immune cell proliferation and differentiation processes, which are jointly regulated by transcription factors (e.g., Runx), post-translational modification (e.g., ubiquitination), and epigenetic regulators (e.g., microRNA) in vertebrates [4,5,6]. Ubiquitination, one of the most fundamental post-translational modifications, regulates most critical cellular processes in eukaryotes [7].

Ubiquitination targets thousands of substrates and governs various cellular processes, including immune cell proliferation and differentiation [8]. Ubiquitin regulation is a cascade reaction involving E1, E2, and E3 enzymes [9]. Among them, E3 ubiquitin ligases, a large and diverse group, are pivotal in the ubiquitination process as they directly bind to the substrates [10]. Based on the structure, three major classes of E3 ubiquitin ligases have been identified, termed as HECT (e.g., E6AP), RING-finger (e.g., Deltex2), and U-box (e.g., CHIP) E3 ligase [11,12,13]. CHIP, also known as STUB1, is an essential E3 ubiquitin ligase that manages the proteolytic turnover of the substrates, involving protein quality control [14]. It has two critical functional regions, the N-terminal tetratricopeptide repeat (TPR) and the highly conserved C-terminal U-box domain [15]. The TPR domain interacts with molecular chaperones like heat shock proteins 70 and 90, while the U-box domain exhibits E3 ligase activity for the ubiquitin–proteasome system-mediated degradation of client proteins [16]. According to research findings, CHIP is known to be widely distributed throughout the cell by targeting various substrates for the efficient degradation of membrane, cytosolic, and nuclear proteins [17].

Studies have indicated that CHIP plays an important role in regulating the proliferation and differentiation of immune cells through interaction with specific transcription factors [18]. CHIP is expressed in hematopoietic stem and progenitor cells, as well as in a variety of immune cells, such as regulatory T (Treg) cells and macrophages [13,19]. Previous studies indicated that CHIP works as a negative regulator of stem cell pluripotency by ubiquitinating and degrading a core set of transcription factors like Runx, SOX2, and NANOG [20]. Runx1 is a member of the Runx transcription factors and plays an important role in hematopoiesis. In another study, it was reported that CHIP directly interacts with Runx1 in the nucleus, leading to its ubiquitination and subsequent degradation, thus suppressing hematopoietic properties [21]. The over-expression of CHIP induced Runx1 ubiquitination and degradation through the ubiquitin–proteasome pathway [22]. CHIP also interacts with other transcription factors like FOXP3, a determinant of Treg cell development and functional maintenance, to promote its K48-linked polyubiquitination and degradation, thereby affecting the suppressive function of Treg cells [23], suggesting the importance of the CHIP-mediated stabilization of core transcription factors in both hematopoiesis and the immune system.

Mollusca is strategically positioned in animal phylogeny, as they first evolve specialized immune cells (namely, haemocytes) responsible for the defense against invading pathogens. Molluscs lack adaptive immunity and instead depend on an innate, non-lymphoid immune system that is carried out by haemocytes [24]. Haemocytes, similar to vertebrate leukocytes with defensive capacities, act as both the undertaker of cellular immunity endowed with phagocytic capacity, encapsulating, and nodule-forming, and the supplier of humoral products, such as microbicidal substances [24,25]. Additionally, molluscs have an ‘open’ circulatory system in which the haemocytes bathe all the organs and tissues, stressing the biological significance of these ancient innate immune cells [24]. However, no typical hematopoietic organs and tissues are found in molluscs as compared to vertebrates [26], and the processes of haemocyte proliferation and differentiation, along with their underlying molecular mechanisms, remain largely unknown.

Among molluscs, the Pacific oysters, a highly prized and prevalent species in aquaculture, serve as a valuable and tractable model for examining the impact of ubiquitination on the proliferation and differentiation of ancient immune cells [27]. In the present study, an E3 ligase CHIP (defined as *Cg*CHIP) was identified and functionally characterized. It was expressed in various haemocyte types and involved in the regulation of haemocyte composition during the immune response, suggesting it might be involved in regulating haemocyte proliferation and differentiation. To further elucidate the specific molecular mechanism process by which *Cg*CHIP knockdown increased the percentage of granulocytes, it was found that *Cg*CHIP directly interacted with the substrate *Cg*Runx, leading to the ubiquitination and subsequent degradation of the *Cg*Runx protein in the haemocyte nucleus. Our findings elucidated that *Cg*CHIP suppressed the proliferation and differentiation of oyster haemocytes by targeting *Cg*Runx for ubiquitination and degradation. These results offered a glimpse into the evolutionary regulation of ubiquitylation in immune cell proliferation and differentiation, which are crucial for immune homeostasis.

## 2. Materials and Methods

### 2.1. Oysters and Mice

All animal experiments were performed according to the animal ethics guidelines approved by the Ethics Committee of the Dalian Ocean University. Adult Pacific oysters (two years old) were collected from an aquaculture farm in Dalian, Liaoning Province, China, and acclimated in filtered, aerated seawater at 20 °C for one week before the following experiments. The oysters were fed with commercial spirulina and seawater was changed daily. Female Kunming mice (eight weeks old) were obtained from the Dalian Medical University for polyclonal antibody preparation. They were acclimated in temperature and humidity-controlled cages for one week, with a 12 h light–dark cycle, and free access to water and rodent food.

### 2.2. Treatment with V. splendidus and Sample Collection

*Vibrio splendidus* strain JZ6 was previously isolated from the lesions of moribund scallop *Patinopecten yessoensis* and preserved in our laboratory [28], and was also confirmed to be the pathogen for oyster *Crassostrea gigas*. It was cultured in 2216E medium at 16 °C for 24 h, and harvested by centrifugation at 5000× *g* for 10 min. After being washed with filtered seawater (0.22 μm filter membrane, Millipore, Bedford, MA, USA), the pellet was resuspended in filtered seawater and adjusted to a final concentration of 1 × 10^9^ CFU/mL for the subsequent stimulation experiment.

For the *V*. *splendidus* stimulation experiment, one hundred and twenty-six oysters were randomly divided into two groups, the seawater (SW) and *V*. *splendidus* (Vs) groups, which individually received an injection with 100 μL of filtered seawater and 100 μL of live *V*. *splendidus* resuspension, respectively. The haemolymph was collected from nine oysters at 0, 3, 6, 12, 24, 48, and 72 h after stimulation, and the haemocytes were harvested by centrifugation at 4 °C, 600× *g* for 10 min. Six tissues, including the hepatopancreas, mantle, gonad, adductor muscle, labial palp, and gill, were collected from the other nine normal oysters. The samples from three oysters were pooled together as one biological replicate, and there were three biological replicates for each tissue and haemocytes at each time point (*n* = 3). All the samples were immediately homogenized in TRIzol reagent (Thermo Fisher Scientific, Waltham, MA, USA) for total RNA extraction.

### 2.3. RNA Interference and Haemocyte Collection

Primers incorporated with the T7 promoter were designed to amplify the cDNA fragments of *Cg*CHIP (P5-P6, Appendix A) and *Cg*Runx (P7-P8, Appendix A) from oyster haemocytes, and EGFP (P9-P10, Appendix A) from the pEGFP vector, respectively. The PCR products served as templates to synthesize double-stranded RNA (dsRNA) via in vitro transcription with HiScribe^®^ T7 High Yield RNA Synthesis Kit (NEB, Ipswich, MA, USA). The integrity of dsRNA was examined using electrophoresis. The concentration of dsRNA was quantified by using Nanodrop 2000 (Thermo Fisher Scientific, Waltham, MA, USA), and standardized to a final concentration of 1 μg/μL in filtered seawater [29].

A total of two hundred and forty-three oysters were divided randomly into four groups for in vivo RNAi experiments. The oysters in the seawater (SW), dsEGFP+Vs, ds*Cg*Runx+Vs, and ds*Cg*CHIP+Vs groups individually received an injection of 100 μL of sterile seawater, and 100 μL of dsRNA of EGFP, *Cg*Runx, and *Cg*CHIP, respectively. The oysters in the latter three groups received another injection of 100 μL of live *V. splendidus* (1 × 10^9^ CFU/mL) at 12 h after dsRNA injection. Haemocytes were collected at 12 h after *V. splendidus* stimulation for total RNA and protein extraction, and flow cytometry analysis. Haemocytes from three individuals were pooled together as one sample. There were three replicates for each assay (*n* = 3).

### 2.4. Gene Cloning and Sequence Analysis

The gene-specific primers *Cg*CHIP-Forward (*Cg*CHIP-F, P1, Appendix A) and *Cg*CHIP-Reverse (*Cg*CHIP-R, P2, Appendix A) were designed according to the sequence of *Cg*CHIP (GenBank accession number: LOC105334775). The PCR product was purified, cloned into a pMD 19-T vector (TaKaRa, Beijing, China), and confirmed through sequencing after it transformed into Trans5α chemically competent cells (Invitrogen, Carlsbad, CA, USA). The functional domain analysis of *Cg*CHIP was conducted with its deduced amino acids via the simple modular architecture research tool (SMART) (https://smart.embl.de/smart/job_status.pl?jobid=112424556139801726077477VQQUNVLpcR). The ubiquitin-modified lysine residues were predicted with UbPreb prediction (http://www.ubpred.org/cgi-bin/ubpred/ubpred.cgi). Multiple sequence alignment of CHIP proteins was constructed with a multiple alignment show program (http://www.bio-soft.net/sms/). An unrooted phylogenetic tree, including CHIP proteins from different species downloaded from NCBI databases, was constructed with Mega 6.0 using the neighbor-joining method. The bootstrap was set at 1000 for reliability branching [30].

### 2.5. Recombinant Expression, Purification, and Polyclonal Antibody Preparation

Specific primer pairs, P3 and P4 (Appendix A), both with *Bam*H I and *Hin*d III sites, were used to amplify the open reading frame fragments of *Cg*CHIP (837 bp). The PCR product was inserted into a pET-28a vector that carried *Bam*H I/*Hin*d III sites. The recombinant plasmids (pET-28a-*Cg*CHIP) were further transformed into BL21 (DE3) pLysS chemically competent cells (TransGen Biotech, Beijing, China). After the cells grew to OD_600_ = 0.5, isopropyl-β-D-thiogalactoside (IPTG) was added at a final concentration of 0.5 mM to induce the expression of recombinant proteins. The *Cg*CHIP recombinant protein (r*Cg*CHIP) was purified through the His-tag Purification Resin (Beyotime, Shanghai, China). The purity and concentration of the r*Cg*CHIP protein were examined using SDS-polyacrylamide gel electrophoresis (SDS-PAGE) and a BCA Protein Assay Kit (Thermo Fisher Scientific, Waltham, MA, USA), respectively. The r*Cg*CHIP and r*Cg*Runx proteins were stored at –80 °C for polyclonal antibody production.

To prepare polyclonal antibodies, six Kunming mice were immunized with the purified r*Cg*CHIP and r*Cg*Runx proteins (1 mg/mL) according to the previous method [31]. Seven days after the fourth immunization, the blood was collected from the immunized mouse, tipped at 4 °C overnight, and centrifuged at 3000 rpm for 10 min to harvest the serum. The specificity of *Cg*CHIP and *Cg*Runx polyclonal antibodies was confirmed by using their recombinant proteins and haemocyte endogenous proteins with a Western blotting assay.

### 2.6. Immunofluorescence Staining

The collected haemocytes were fixed with 4% paraformaldehyde (Thermo Fisher Scientific, Waltham, MA, USA) for 15 min, permeabilized with 0.1% Triton X-100 for 10 min, and blocked with 3% BSA at 37 °C for 1 h. Following that, haemocytes were incubated with primary antibodies against *Cg*CHIP (1:500) and *Cg*Runx (1:1000) at 4 °C for a whole night, respectively. The following day, they were incubated with a secondary antibody conjugated to Alexa Fluor 488 (1:1000, Abcam, Cambridge, UK) at 37 °C for 1 h. After the final three times of washing with TBST, the glasses were mounted with ProLong Glass Antifade Mountant containing NucBlue Stain (Thermo Fisher Scientific, Waltham, MA, USA). A laser confocal microscope (LSM 800, ZEISS, Oberkochen, Germany) was used to take the images.

### 2.7. Co-Immunoprecipitation Assay

Proteins were extracted from the haemocytes using an immunoprecipitation lysis buffer containing a protease inhibitor mixture (Solarbio life sciences, Beijing, China), followed by overnight incubation with 1 μg of the antibody against *Cg*CHIP (1:50). The next day, proteins were incubated with Protein A/G Magnetic Beads (Thermo Fisher Scientific, Waltham, MA, USA) at room temperature for 2 h. After washing with the immunoprecipitation buffer, the *Cg*CHIP antibody binding proteins were examined with *Cg*CHIP and *Cg*Runx antibodies by Western blotting, respectively [32].

### 2.8. In Vitro Ubiquitination Assay

The in vitro ubiquitination assay was performed to determine the Ub-protein ligase activity of *Cg*CHIP on *Cg*Runx according to a previous study [33]. The reaction volume was 20 μL, containing 100 nM E1 (UBPBio, Dallas, TX, USA), 2 mΜ E2 (UBPBio, Dallas, TX, USA), 2 μM r*Cg*CHIP, 2 μM r*Cg*Runx, 50 μM Ub (UBPBio, Dallas, TX, USA), 2 mM ATP (UBPBio, Dallas, TX, USA), and 1 μL of glycerol as well as 2 μL of 10 × ubiquitination buffer (10 mM Tris-HCl pH 7.5, 50 mM NaCl, 10 mM βME, and 5 mM MgCl_2_). The reaction without r*Cg*CHIP or r*Cg*Runx was employed as the negative control, while Ub was absent and set as the blank group. After incubation at 37 °C for 2 h, the reaction was stopped by adding 5 μL of protein loading buffer, and the mixture was finally examined by Western blotting with the Ub polyclonal antibody.

### 2.9. Molecular Docking

*Cg*CHIP and *Cg*Runx were obtained from the UniProt website, and their three-dimensional structures were predicted by AlphaFold. Additionally, PyMOL (version 2.5) was applied to evaluate and visualize the interactions between the two proteins [32].

### 2.10. Quantitative Reverse Transcription-Polymerase Chain Reaction (qRT-PCR)

The total RNA was extracted from haemocytes and tissues using the TRIzol reagent following the manufacturer’s protocol, and cDNA was synthesized from RNA using a TransScript one-step gDNA removal and cDNA synthesis kit (TransGen Biotech, Beijing, China). qRT-PCR was carried out with PrimeScript^TM^ RT Master Mix using Light Cycler 7500 Real-Time PCR System (Applied Biosystems^®^, Carlsbad, CA, USA), and it normalized the expression levels to *Cg*EF-1α. The relative expression levels of target genes were analyzed with the 2^−ΔΔCT^ method. The corresponding primer sequences used for qRT-PCR are listed in Appendix A.

### 2.11. Protein Abundance Quantitation with Western Blottting

The total proteins from haemocytes were extracted using RIPA lysis buffer as described previously [32]. SDS-PAGE was used to separate the proteins, and then they were transferred to nitrocellulose membranes via a mini-transfer tank for electrophoresis. After incubation in blocking buffer with 3% BSA at 37 °C for 3 h, the membranes were incubated with primary antibodies at 4 °C overnight, followed by incubation with HRP-conjugated goat anti-rabbit or goat anti-mouse IgG secondary antibodies (1:1000, Proteintech, Chicago, IL, USA) at 37 °C for 1 h. After they were heavily washed three times with TBST, the membranes were finally incubated with SuperSigna ECL Western blot substrates for 30 s, and imaged with Amersham Imager 600.

The mouse polyclonal antibodies against *Cg*CHIP (1:1000), *Cg*Runx (1:1000), *Cg*Integrin α4 (1:2000), and *Cg*AATase (1:1000) were prepared previously in our laboratory. The other antibodies used for Western blotting were monoclonal antibodies, including rabbit monoclonal antibodies against PCNA (1:1000, Cell Signaling Technology, Boston, MA, USA), rabbit monoclonal antibodies against Cyclin B1 (1:1000, ABclonal, Wuhan, China), rabbit monoclonal antibodies against CDK2 (1:1000, Cell Signaling Technology, Boston, MA, USA), mouse monoclonal antibodies against Histone H3 (1:5000, Proteintech, Chicago, IL, USA), and HRP-conjugated rabbit monoclonal against beta-Tubulin (1:10,000, Proteintech, Chicago, IL, USA).

### 2.12. The Sorting and Composition Change of Haemocytes with Flow Cytometry Analysis

Haemocytes collected from nine oysters in the SW+SW, dsEGFP+Vs, ds*Cg*CHIP+Vs, and ds*Cg*Runx+Vs groups were immediately fixed with a 4% paraformaldehyde reagent at room temperature for 15 min. Following that, they were mounted for detection with flow cytometry. Agranulocyte, semi-granulocyte, and granulocyte subpopulations of haemocytes were gated and divided according to their size (forward scatter, FSC), internal complexity (side scatter, SSC), and percentage, referring to a previous study [34]. The percentage change in three subpopulations in total haemocytes was analyzed by flow cytometry.

### 2.13. Isolation and Cultivation of Agranulocytes In Vitro

The collected haemocytes from nine oysters in the SW, dsEGFP, ds*Cg*CHIP, and ds*Cg*Runx groups were resuspended in the modified Alsever’s solution and the concentration was adjusted to 10^7^ cells/mL. In total, 2 mL of haemocyte resuspension was layered onto the top of the Percoll density gradient composed of 30%/55% Percoll. After density gradient centrifugation at 600× *g* for 15 min, agranulocytes in the upper layer of the 30%/55% Percoll gradient (Sigma-Aldrich, Saint Louis, MO, USA) were extracted and collected. They were adjusted to 10^6^ cells/mL and plated in 6-well plates with 2 mL of modified L-15 medium at 20 °C for in vitro culturing. The medium was replaced every two days by replacing 1/4 of the spent medium with an equal volume of fresh medium that was pre-warmed at 20 °C. After treatment with 2 μg/mL of Lipopolysaccharide (LPS) (*Eschrichia coli* LPS 0111: B4, Sigma-Aldrich, Saint Louis, MO, USA) for 24 h, the haemocytes cultured in vitro were collected at 7th day and mounted for detection with flow cytometry. The percentage of semi-granulocyte and granulocyte subpopulations that differentiated from the agranulocyte subpopulation was examined with flow cytometry. The differentiation of haemocytes was represented by the percentage change in semi-granulocyte and granulocyte subpopulations in total haemocytes [35].

### 2.14. The New-Born Haemocyte Observation with EdU Labelling

EdU labelling assay was performed to detect the new-born cells in agranulocytes and granulocytes using Click-iT Plus EdU Alexa Fluor 488 Flow Cytometry Assay Kit (Thermo Fisher Scientific, Waltham, MA, USA), according to the manufacturer’s instruction, and examined with flow cytometry (BD Biosciences, Franklin Lake, NJ, USA). The proliferation of agranulocytes and granulocytes was defined by the new-born (EdU^+^) agranulocytes and granulocytes, respectively. The percentage of EdU^+^ agranulocytes and granulocytes in total agranulocytes and granulocytes was examined and analyzed with flow cytometry, respectively.

### 2.15. Determination of Cell Cycle Phase Proportion of Haemocytes by Flow Cytometry

The cycle phase of haemocytes was detected by flow cytometry to analyze the proliferation activity of haemocytes. The collected haemocytes were fixed in 70% ethanol at –20 °C overnight. After two washes with cold PBS, the haemocytes were incubated with RNaseA (Beyotime, Shanghai, China) and propidium iodide (Beyotime, Shanghai, China) in the dark at 37 °C for 30 min, and then mounted for detection with flow cytometry. The phases of the cell cycle were analyzed with ModFit LT 5.0 (BD Biosciences, Franklin Lake, NJ, USA). The cell cycle was divided into four distinct phases, G1 (gap 1), S (DNA synthesis), G2 (gap 2), and M (mitosis). The DNA content of cells in the G2 and M phases was twice that of cells in the G1 and G0 phases. The proliferation of haemocytes was represented by the percentage of haemocytes in the G1/G0, S, and G2/M phases in the total haemocytes.

### 2.16. Determination of Haemocyte Phagocytosis with Flow Cytometry

Haemocytes (10^6^ cells/mL) from nine oysters in the SW, dsEGFP+Vs, ds*Cg*CHIP+Vs, and ds*Cg*Runx+Vs groups were collected and incubated with 5% red-labeled latex beads (10^8^ beads/mL, 2 μm diameter, Sigma-Aldrich, Saint Louis, MO, USA) at room temperature for 1 h. Extracellular fluorescence was quenched by adding 10% trypan blue. Following that, haemocytes were mounted for detection with flow cytometry. Phagocytic capacity of haemocytes was defined by their phagocytic rate. Phagocytic rate of haemocytes was measured by calculating the percentage of PE^+^ haemocytes that engulfed latex beads in total haemocytes [34].

### 2.17. Statistical Analysis

Triplicate replications (*n* = 3) were performed for each experiment. Data were graphed and analyzed using Origin 8.1 (OriginLab, Northampton, MA, USA) and Statistical Package for Social Sciences (SPSS) 16.0. Statistical differences between two groups were assessed using the Student’s *t*-test. A one-way analysis of variance test (ANOVA) followed by Dunnett’s post detection were used to compare differences among multiple groups.

## 3. Results

### 3.1. Oysters Possess a Typical E3 Ligase CHIP

The complete open reading frame of *Cg*CHIP was 837 bp, encoding a peptide of 278 amino acids with a molecular mass of 32.4 kDa. It comprised three regions, an N-terminal triple tetratricopeptide repeat (TPR) domain, a low complexity domain, and a C-terminal U-box domain (Figure 1A). The same region arrangement was observed in the human homolog (Figure 1B). Particularly, oyster CHIP shared a high similarity in tertiary structure with human CHIP, as predicted by the SWISS-MODEL (Figure 1A,B). Multiple sequence alignment showed that *Cg*CHIP shared 67–97% similarity with its paralogous from other molluscs including *Crassostrea virginica* (XP_22319432.1), *Mizuhopecten yessoensis* (XP_021380132.1), *Aplysia californica* (XP_005108643.1), and *Biomphalaria glabrata* (XP_013068280.1), and a 62% similarity with its homologous from *Drosophila melanogaster* (NP_47744.1) (Figure 1C). All the CHIPs were separated into two distinct clades of vertebrates and invertebrates. *Cg*CHIP was clustered with *Cv*CHIP from *C. virginica*, and then assigned to the invertebrate clade in the phylogenetic tree (Figure 1D). The results indicated that CHIP was evolutionarily conserved.

### 3.2. CgCHIP Is Highly Expressed in Oyster Haemocytes

The mRNA levels of *Cg*CHIP in haemocytes and tissues were examined by qRT-PCR. Its mRNA transcripts were found to be ubiquitously expressed in haemocytes and six tested tissues, with the highest level in haemocytes, which was 6.72-fold (*p* < 0.01) of that in the hepatopancreas (Figure 2A). Its temporal expression levels in haemocytes after *V. splendidus* stimulation were further detected by qRT-PCR. The mRNA level of *Cg*CHIP in haemocytes increased and peaked at 3 h in the Vs group, which was 1.37-fold (*p* < 0.05) of that in the SW group (Figure 2B). Following this, there was a decrease in its mRNA levels in haemocytes from 6 to 72 h, with notable changes detected at 6 and 72 h, which was 0.53-fold (*p* < 0.05) and 0.44-fold (*p* < 0.01) of that in the SW oysters (Figure 2B). The higher expression in haemocytes and the alterations post-stimulation suggested that *Cg*CHIP might play a role in the cellular processes and immune responses of oyster haemocytes.

### 3.3. CgCHIP Is Widely Distributed throughout the Haemocytes

The recombinant protein of *Cg*CHIP was expressed and purified with the His-tag Protein Fusion and Purification System. A distinct band of r*Cg*CHIP with a molecular weight of 34 kDa was revealed by SDS-PAGE (Figure 2C), which was consistent with the predicted molecular weight of *Cg*CHIP (32.4 kDa) with an “HHHHHH” His-tag (about 1.6 kDa), respectively. The anti-*Cg*CHIP polyclonal antibody was prepared by immunizing mice using the recombinant proteins, and it could recognize both the corresponding recombinant protein and natural protein from haemocytes without any non-specific binding, as shown with Western blotting, confirming the high specificity of the *Cg*CHIP antibody (Figure 2D).

The circulating haemocytes were collected from oyster haematocoel (Figure 2F) and comprised morphologically and functionally heterogeneous cell populations (Figure 2F,G). Three subpopulations of haemocytes were morphologically identified and separated as agranulocytes (A), semi-granulocytes (SG), and granulocytes (G) by flow cytometry according to our previous report [34]. Next, immunofluorescence staining confirmed that *Cg*CHIP was widely expressed in the three haemocyte subpopulations. The positive signals of *Cg*CHIP labeled with Alexa Fluor 488 were visible in green, and widely distributed in the cytoplasm and nucleus throughout entire haemocytes (Figure 2I). The mean fluorescence intensity of *Cg*CHIP was higher in granulocytes, which was approximately 1.26-fold (*p* < 0.05) and 2.67-fold (*p* < 0.01) of that in semi-granulocytes and agranulocytes (Figure 2J), which was well consistent with transcriptome data (Figure 2E). Fifty cells of each haemocyte subpopulation were counted and quantified using ImageJ software (*n* = 50). These results showed that *Cg*CHIP was ubiquitously expressed in the three subpopulations of haemocytes and widely distributed throughout the cells, suggesting it could potentially affect multiple substrates and control various cellular processes in oyster haemocytes.

### 3.4. CgCHIP Alters the Proportion of Three Haemocyte Subpopulations

To investigate the effect of *Cg*CHIP on the cellular processes of haemocytes, the percentage of the three haemocyte subpopulations was determined by flow cytometry following the knockdown of *Cg*CHIP with its dsRNA. Compared to the dsEGFP+Vs group, the percentage of agranulocytes in total haemocytes reduced from 34.2% to 21.7% in the ds*Cg*CHIP+Vs oysters, which was 0.63-fold of that in the dsEGFP+Vs group (Figure 2K,L). The percentage of granulocytes in total haemocytes increased from 26.3% to 32%, which was 1.22-fold of that in the dsEGFP+Vs group (Figure 2K,L). Next, the effect of *Cg*Runx on the cellular processes of haemocytes was examined following *Cg*Runx knockdown with its dsRNA. The knockdown efficiency of *Cg*Runx in haemocytes was confirmed at the protein level by Western blotting. Compared with the dsEGFP+Vs group, the protein expression level of CgRunx in the dsCgRunx+Vs group was decreased significantly (Appendix A). On the contrary, there was an increased percentage of agranulocytes from 34.2% to 46.3% and a decreased percentage of granulocytes from 26.3% to 17.4% in the ds*Cg*Runx+Vs oysters, compared with the dsEGFP+Vs oysters (Figure 2K,L). These results showed that suppressing *Cg*CHIP expression resulted in a lower proportion of agranulocytes and a higher proportion of granulocytes, whereas reducing *Cg*Runx expression had the opposite effects, suggesting *Cg*CHIP functioned as a negative regulator in the proliferation and differentiation of haemocytes during the immune response.

### 3.5. CgCHIP Targets CgRunx Protein

It was shown that CHIP could directly target Runx as predicated via bioinformatics (Figure 3A). The *Cg*Runx protein might have a potential Met-1 ubiquitination site located at the N-terminal, and four potential Lys ubiquitination sites (K65, K107, K126, and K149) clustered in its Runt domain (Figure 3B). Next, an immunoprecipitation experiment with haemocyte lysate (Appendix A) was performed to obtain the *Cg*CHIP antibody binding proteins, and these binding proteins were further examined with anti-*Cg*CHIP antibodies and anti-*Cg*Runx antibodies by Western blotting, respectively. Anti-*Cg*Runx antibodies could recognize the *Cg*CHIP antibody binding protein with an immunoreactive band of about 60 kDa, suggesting *Cg*CHIP interacted with the *Cg*Runx protein (Figure 3C). To further investigate the specific regions through which the two proteins interact, molecular docking analysis was conducted, and it was found that the *Cg*CHIP fragment containing amino acids 147–270 might bind to the *Cg*Runx fragment containing amino acids 91–435 (Figure 3D,E).

Next, the ubiquitination and degradation activity of the *Cg*CHIP protein was determined using a ubiquitination assay in vitro with commercial Ub-activating enzyme E1, Ub-conjugating enzyme E2, and Ub in the ubiquitination buffer. Two distinct bands corresponding to *Cg*CHIP and Ub were revealed by Western blotting (Figure 3F), indicating the ubiquitination activity of the *Cg*CHIP protein. The conjugation reaction between *Cg*CHIP and its potential substrate *Cg*Runx yielded a large number of ubiquitination products in the form of a ladder (Figure 3G, lane 1), while no significant smear or ladder was observed in the control groups (Figure 3G, lane 3–4). Moreover, the protein content of *Cg*Runx was restored by the addition of MG132, a proteasome inhibitor (Figure 3H). These results suggest that *Cg*CHIP could directly interact with the *Cg*Runx protein and mediate its ubiquitination and degradation.

### 3.6. CgCHIP Enhances the Ubiquitination and Degradation of CgRunx

*Cg*Runx was identified in our previous study and proven to be involved in haematopoiesis and immune responses in oysters [29]. Following that, its expression profile was analyzed in the three subpopulations of haemocytes using transcriptome data analysis and immunofluorescence staining, and it was found to be widely expressed in all three subpopulations. Immunofluorescence observed that *Cg*Runx was localized in the nucleus of haemocytes (Figure 4A). The mean fluorescence intensity of *Cg*Runx was higher in semi-granulocytes, which was approximately by 1.11- and 1.31-fold (*p* < 0.05) than in agranulocytes and granulocytes (Figure 4C), which accurately matched the transcriptome data (Figure 4B).

To investigate the effect of *Cg*CHIP on *Cg*Runx protein levels, an RNAi of *Cg*CHIP was performed by an injection with dsRNA (Figure 4D), and results revealed that the knockdown of *Cg*CHIP increased the protein levels of *Cg*Runx. The knockdown efficiency of *Cg*CHIP in haemocytes was first confirmed at the mRNA and protein levels by qRT-PCR and Western blotting analysis (Figure 4E–G). Compared with the dsEGFP+Vs group, the mRNA and protein expression levels of *Cg*CHIP in the ds*Cg*CHIP+Vs group were both decreased (Figure 4E–G), which were 0.36-fold (*p* < 0.05) and 0.58-fold (*p* < 0.05) of that in the dsEGFP+Vs group (Figure 4E–G), respectively. Meanwhile, the protein level of *Cg*Runx in ds*Cg*CHIP oyster haemocytes was found to increase at 12 h after *V*. *splendidus* stimulation, which was 1.68-fold (*p* < 0.05) of that in the dsEGFP+Vs group (Figure 4F,G). These results suggest that *Cg*CHIP mediated the degradation of the *Cg*Runx protein via ubiquitination.

### 3.7. CgCHIP Negatively Regulates the Differentiation of Agranulocytes

Agranulocytes in the oyster immune system are a group of precursor haemocytes that are undifferentiated and immature, differentiating into semi-granulocytes and eventually into granulocytes. To examine the function of *Cg*CHIP in the regulation of agranulocyte differentiation, agranulocytes were isolated from the total haemocytes that knocked down *Cg*CHIP and were cultured in vitro (Figure 5A). Agranulocytes underwent differentiation into semi-granulocytes or granulocytes after stimulation with LPS to examine their differentiation at 7 d (Figure 5A). The semi-granulocytes and granulocytes were located outside the higher-density cluster of agranulocytes in the flow cytometry volcano plots with higher FSC and SSC values (Figure 5B). The differentiation of agranulocytes was defined by the production of semi-granulocytes and granulocytes. There was a 2.4% increase (from 10.0% in dsEGFP+LPS oysters to 12.4% in ds*Cg*CHIP+LPS oysters) and a 5.1% decrease (from 10.0% in dsEGFP+LPS oysters to 4.9% in ds*Cg*Runx+LPS oysters) in the percentage of semi-granulocytes and granulocytes differentiated from agranulocytes at 7 d with LPS stimulation for 24 h (Figure 5B,C).

Meanwhile, the protein abundance of PCNA (a molecular marker for cell proliferation), Integrin α4 (a molecular marker for agranulocytes), and AATase (a molecular marker for granulocytes) in the mixed cells derived from the primary culture agranulocytes at 7 d was determined to support the inhibitory role of *Cg*CHIP in agranulocyte differentiation (Figure 5D). The abundance of *Cg*PCNA and *Cg*Integrin α4 both decreased significantly, which was 0.39-fold (*p* < 0.01) and 0.39-fold (*p* < 0.01) of that in the dsEGFP+LPS oysters, respectively (Figure 5D,E), whereas the protein abundance of *Cg*AATase increased significantly (1.57-fold, *p* < 0.05, Figure 5D,E). These results indicate that *Cg*CHIP negatively regulated the differentiation of agranulocytes by targeting the hematopoietic transcription factor *Cg*Runx in oyster *C*. *gigas*.

### 3.8. CgCHIP Negatively Regulates the Proliferation of Granulocytes

Moreover, granulocyte proliferation could be another key factor contributing to its increase in proportion. Therefore, the proliferation of agranulocytes and granulocytes was detected after *Cg*CHIP was knocked down by flow cytometry. It was found that the knockdown of *Cg*CHIP increased granulocyte proliferation but did not influence agranulocyte proliferation. Agranulocytes and granulocytes were separated into two populations according to their FSC and SSC values in the scatter plots of flow cytometry. The percentage of agranulocytes with EdU labeling in the ds*Cg*CHIP+Vs oysters was 3.53%, which increased slightly (0.44%) compared to that in the dsEGFP+Vs oysters, and there was no significant difference (*p* > 0.05, Figure 6A,B), whereas the percentage of granulocytes with EdU labeling was 3.76% in the ds*Cg*CHIP+Vs oysters, which was significantly higher than that in the dsEGFP+Vs (2.09%) and seawater oysters (1.95%, *p* < 0.05, Figure 6C,D).

To further explore the effects of *Cg*CHIP and *Cg*Runx on granulocyte proliferation, granulocytes were isolated from the total haemocytes that knocked down *Cg*CHIP or *Cg*Runx for the cell cycle distribution analysis (Figure 6E). The percentage of granulocytes in the G0/G1 and G2/M phases after the knockdown of *Cg*CHIP reduced from 58.1% to 54.3% (0.93-fold, *p* < 0.05) and increased from 13.0% to 17.3% (1.33-fold, *p* < 0.05) with *V. splendidus* stimulation, compared to the dsEGFP+Vs oysters (Figure 6F,G). However, there was no remarkable change in the S phase of granulocytes (Figure 6F,G). These results showed that the knockdown of *Cg*CHIP increased the G2/M phase of granulocytes. Meanwhile, the effect of *Cg*Runx on the cell cycle of granulocytes was also measured by flow cytometry. The percentage of granulocytes in the G0/G1 and G2/M phases after the knockdown of *Cg*Runx both reduced from 58.1% to 47.2% (0.81-fold, *p* < 0.01) and from 13.0% to 7.32% (0.56-fold, *p* < 0.05) with *V. splendidus* stimulation. In contrast, the percentage of granulocytes in the S phase in the ds*Cg*Runx+Vs oysters increased remarkably from 28.9% to 48.2% (1.67-fold, *p* < 0.01, Figure 6F,G). The findings indicated that *Cg*CHIP suppressed the proliferation of granulocytes, whereas *Cg*Runx enhanced it.

Next, the protein expression levels of cell cycle marker genes Cyclin B1 and CDK2 in the granulocytes of *C. gigas* post-injection with *Cg*CHIP dsRNA were assessed to corroborate the suppressive function of *Cg*CHIP in granulocyte proliferation. The band intensity of *Cg*Cyclin B1 and *Cg*CDK2 proteins in the ds*Cg*CHIP+Vs oysters both increased compared to that in the dsEGFP+Vs oysters (Figure 6H), which was 1.45-fold (*p* < 0.05) and 1.79-fold (*p* < 0.01) of that in the dsEGFP+Vs oysters (Figure 6I), respectively. These results suggest that *Cg*CHIP negatively regulated the proliferation of granulocytes by targeting the hematopoietic transcription factor *Cg*Runx in oyster *C*. *gigas*.

### 3.9. CgCHIP Negatively Regulates Haemocyte Function through a CgRunx-Dependent Manner in the Terminal Effect

Granulocytes are at the terminal stage of haemocyte differentiation and endowed with the strongest immune competence, displaying excellent phagocytic capability. This research confirmed that the increase in granulocyte number resulted from both their proliferation and differentiation from agranulocytes controlled by *Cg*CHIP in a *Cg*Runx-dependent manner, both essential processes in the immune response against invading pathogens. Next, following the knockdown of *Cg*CHIP and *Cg*Runx, flow cytometry was used to evaluate the phagocytic capability of haemocytes in response to *V. splendidus* stimulation. After incubation with red fluorescent beads, the phagocytic rate of haemocytes in the ds*Cg*CHIP+Vs oysters increased (22.9%), which was 1.17-fold (*p* < 0.05) of that in the dsEGFP+Vs oysters (Figure 7A,B). In contrast, the phagocytic rate of haemocytes in the ds*Cg*Runx+Vs oysters was found to reduce from 19.5% to 14.3% (0.73-fold, *p* < 0.05) compared with that in the dsEGFP+Vs oysters (Figure 7A,B). These findings imply that *Cg*CHIP played an important role in inhibiting haemocyte proliferation and differentiation driven by *Cg*Runx, thus influencing the functional status of haemocytes to maintain homeostasis in oysters.

## 4. Discussion

The proliferation and differentiation of immune cells are an important part of the background machinery for ensuring the proper function of immune cells and the immune system [1]. Importantly, E3 ubiquitin ligases are closely related to cell division and differentiation in all eukaryotes and play crucial regulatory roles in almost all biological processes, such as proliferation and differentiation [7]. In invertebrates, there is still no systematic knowledge about the hematopoietic tissues and hematopoiesis [24,26], partly due to a lack of understanding about the haemocyte proliferation and differentiation process [26]. In the present study, the roles of E3 ubiquitin ligase *Cg*CHIP on the proliferation and differentiation of haemocytes and its underlying mechanism were investigated in oyster *C*. *gigas*, which would provide a new evolution perspective for the ubiquitination, proliferation, and differentiation of immune cells.

CHIP is an evolutionarily conserved protein, belonging to the U box-containing E3 ubiquitin ligases, which is widely found in various species [36]. It contains a U-box domain for E3 ubiquitin ligase activity, a TPR domain responsible for chaperone binding, and a charged domain rich in charged residues [37]. In the present study, a CHIP homolog (*Cg*CHIP) was identified from oysters with a TRR domain at the N-terminal and a U-box domain at the C-terminal. Although there were some slight variations in the amino acid sequence of the TRR domain, the U-box domain was highly conserved in *Cg*CHIP, and it shared similar spatial structures, especially functional domains, compared to its homologs from vertebrates and other invertebrates, suggesting that *Cg*CHIP was a typical U-box E3 ligase enzyme. After incubation with Ub, E1, E2, and ATP, the r*Cg*CHIP+Ub complex was detected with a Ub antibody by an in vitro ubiquitination assay, indicating that *Cg*CHIP possessed the typical U-box E3 ligase activity to trigger the ubiquitination cascade in oysters. Furthermore, immunofluorescence staining showed that *Cg*CHIP was present in a dotted pattern throughout the cytoplasm and nucleus of the entire haemocyte, suggesting it served as E3 ligase-targeted multiple-substrate proteins that function in cellular activities, such as transcriptional regulation.

In the present study, it was hypothesized for the first time that *Cg*CHIP might target *Cg*Runx for ubiquitin-mediated proteasome degradation in oyster haemocytes similar to that in vertebrates [22,37]. CHIP has been reported to interact physically with Runx through its TRR domain in the nucleus and serves as an E3 ubiquitin ligase to regulate Runx1 protein stability via a ubiquitination and degradation mechanism [22,37]. In this study, a typical Met-1 ubiquitination site was found to be located at the N-terminal, and four typical Lys-65, Lys-107, Lys-126, and Lys-149 ubiquitination sites were clustered in the conserved Runt domain of *Cg*Runx. Moreover, the bioinformatics analysis indicated that *Cg*Runx was a potential target gene of *Cg*CHIP. An in vitro ubiquitination assay with their purified recombinant proteins was performed to examine the ubiquitination of *Cg*Runx mediated by *Cg*CHIP. When E1, E2, r*Cg*CHIP, and ATP were added to the reaction system containing purified r*Cg*Runx, a strong, high molecular mass smear was observed while no smear appeared when r*Cg*CHIP was removed from the reaction mixture, indicating that *Cg*CHIP functioned as an E3 ubiquitin ligase for *Cg*Runx. Moreover, the protein content of *Cg*Runx was restored by the addition of a proteasome inhibitor MG132. These results indicate that *Cg*CHIP targeted and ubiquitinated *Cg*Runx leading to its proteasome-mediated degradation and functional inactivation.

Transcription factor Runx is a master regulator for the specification of hematopoietic lineage during embryogenesis, and bone marrow and extramedullary hemopoiesis [38]. In vertebrates, there are three Runx genes (Runx1, Runx2, and Runx3), which share a high degree of functional redundancy in several cellular processes, such as proliferation and differentiation. Runx1 and Runx3 are well known for their essential roles in hematopoiesis and immunity, and they are expressed in almost all adult blood lineages except for erythrocytes [39], whereas only one Runx homolog (*Cg*Runx) has been identified in the oyster genome, and it has been suggested to be involved in immune responses and larvae hematopoiesis [40]. In the present research, it was noted that *Cg*Runx was evenly expressed in the three subpopulations of haemocytes, including the undifferentiated pluripotent agranulocytes, intermediate semi-granulocytes, and terminally differentiated effector granulocytes (Figure 2H) [35], suggesting it played distinct potential roles in different haemocyte subpopulations in a cell-type-specific manner [41]. After the expression of *Cg*CHIP was knocked down by RNAi, the percentage of agranulocytes in total haemocytes decreased while their differentiation activity enhanced. Meanwhile, the percentages of granulocytes in total haemocytes and the G2/M phase, as well as their proliferation activity and the phagocytic rate of haemocytes, all increased. In contrast, after the transcriptional expression of *Cg*Runx was knocked down, the percentage of agranulocytes in total haemocytes increased, the percentage of granulocytes and semi-granulocytes differentiated from the agranulocytes decreased, and the percentage of granulocytes in total haemocytes and in the G2/M phase both decreased. These results indicate that *Cg*CHIP negatively regulated agranulocyte differentiation and granulocyte proliferation by targeting the hematopoietic transcription factor *Cg*Runx in oyster *C*. *gigas*.

Furthermore, several molecular markers of cell proliferation (PCNA), cell cycle (Cyclin B1, CDK2), agranulocyte (*Cg*Integrin α4), and agranulocyte differentiation (*Cg*AATase) were selected and examined to indicate the proliferation and differentiation of haemocytes, respectively. In the present study, the protein contents of *Cg*PCNA and *Cg*Integrin α4 in agranulocytes that were cultured in vitro and stimulated with LPS for 7 d decreased, while *Cg*AATase increased. These results suggest that agranulocytes underwent a differentiation process with LPS stimulation, which was revealed by the protein expression level changes in molecular markers. Meanwhile, the *Cg*Cyclin B1 and *Cg*CDK2 protein contents in granulocytes increased in the ds*Cg*CHIP+Vs oysters, revealing a proliferation process in granulocytes with *V*. *splendidus* stimulation. Ultimately, inhibiting *Cg*CHIP gene expression was found to enhance the increase in granulocyte numbers, supporting granulopoiesis and enabling granulocytes to carry out phagocytosis in oysters.

In conclusion, *Cg*CHIP, an evolutionarily conserved U-box E3 ubiquitin ligase enzyme, was first characterized in oysters, and its function and underlying mechanism in haemocyte metamorphosis were elucidated. *Cg*CHIP targeted and ubiquitinated *Cg*Runx, leading to proteasome-mediated degradation and thus, the functional inactivation of *Cg*Runx. Then, *Cg*CHIP was proven to prevent agranulocyte differentiation and granulocyte proliferation through its inhibition of *Cg*Runx transcriptional activity (Figure 8). Altogether, these results revealed the novel functional properties of *Cg*CHIP in orchestrating agranulocyte differentiation and granulocyte proliferation and delineated a conserved role of *Cg*CHIP as a negative regulator of the hematopoietic master regulator, *Cg*Runx protein stability in oyster *C*. *gigas*.

## Figures and Tables

**Figure 1 cells-13-01535-f001:**
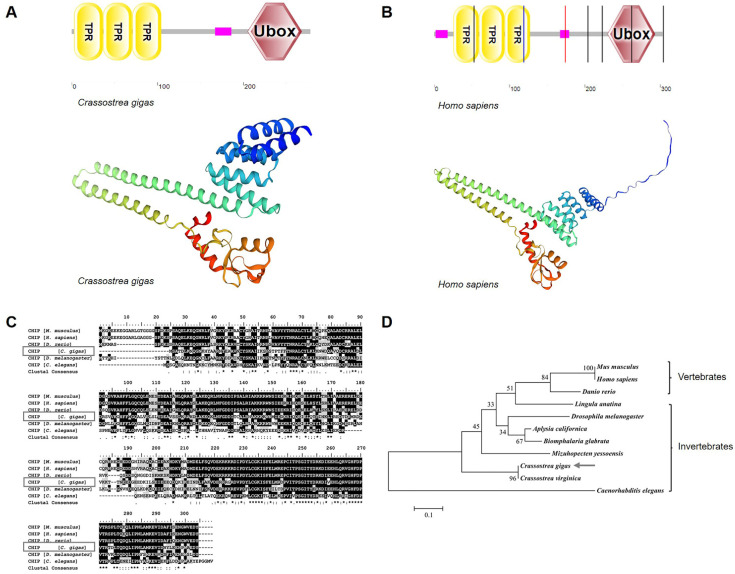
Evolutionary properties of ubiquitin E3 ligase CHIP from oyster *C*. *gigas*. (**A**) Domain and tertiary structure prediction of CHIP from oyster *C***.**
*gigas* by SMART and SWISS-MODEL program. (**B**) Domain and tertiary structure prediction of CHIP from *Homo sapiens* by SMART and SWISS-MODEL program. The pink box indicates a low complexity domain. (**C**) Multisequence alignment analysis of *Cg*CHIP with its homologues from other vertebrate and invertebrate species. Amino acids with 100% identity are in black, and similar amino acids are in gray. (**D**) A phylogenetic tree for CHIP was constructed with the amino acid sequences from the indicated species including *H. sapiens*, *M. musculus*, *D. rerio*, *L. anatine*, *D. melanogaster*, *A. californica*, *B. glabrata*, *M. yessoensis*, *C*. *gigas*, *C. virginica*, and *C. elegans*. The trees were constructed using the neighbor-joining (NJ) algorithm in the Mega 6.0 program based on multiple sequence alignment by ClustalW. Bootstrap values of 1000 replicates (%) are indicated for the branches. CHIP from *C*. *gigas* was marked with a grey arrow.

**Figure 2 cells-13-01535-f002:**
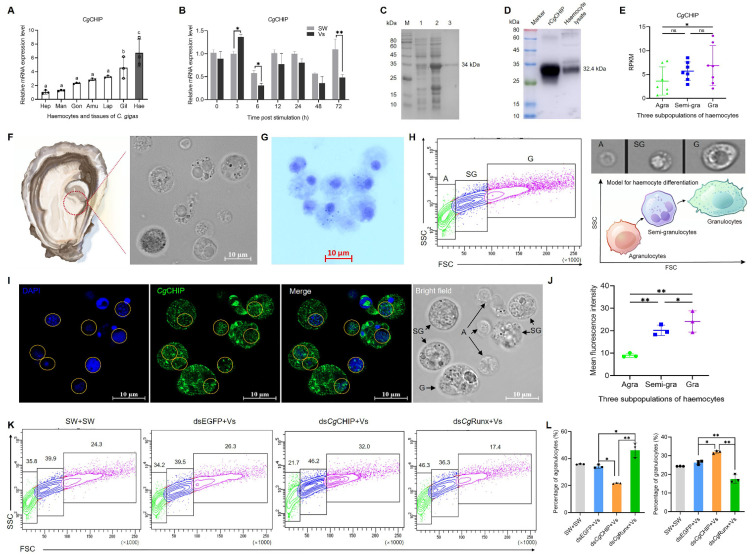
*Cg*CHIP is highly expressed in oyster haemocytes and alters the proportion of their three subpopulations. (**A**) The mRNA transcripts of *Cg*CHIP in the indicated tissues and haemocytes examined by qRT-PCR, normalized to *Cg*EF1-α. Hep: hepatopancreas; Man: mantle; Gon: gonad; Amu: adductor muscle; Lap: labial palp; Gil: gill; Hae: haemocytes. *p*-values, ^a^ *p* > 0.05, ^b^
*p* < 0.05, and ^c^
*p* < 0.01, were calculated using a one-way ANOVA with Dunnett’s correction for multiple comparisons. (**B**) Relative temporal levels of *Cg*CHIP mRNA in haemocytes with or without *V. splendidus* infection examined by qRT-PCR, normalized to *Cg*EF1-α. Error bars show mean ± standard deviation. *p*-values, * *p* < 0.05, ** *p* < 0.01, were calculated using a two-tailed, unpaired *t*-test. Error bars show mean ± standard deviation (*n* = 3). (**C**) SDS-PAGE analysis showed the recombinant-*Cg*CHIP (r*Cg*CHIP) proteins. Lane M: protein molecular marker; Lane 1: negative control (without IPTG induction); Lane 2: induced recombinant protein with IPTG; Lane 3: purified r*Cg*CHIP protein. (**D**) The specificity of the *Cg*CHIP polyclonal antibody determined by Western blotting. Lane M: protein molecular marker; Lane 1: in vitro recombinant proteins; Lane 2: haemocyte lysate. (**E**) Transcriptome data analysis shows the mRNA transcripts of *Cg*CHIP in the three haemocyte subpopulations (*n* = 7). *p*-values, * *p* < 0.05, were calculated using a one-way ANOVA with Dunnett’s correction for multiple comparisons. ns indicates no significant difference. (**F**) Haemocytes collected from oyster haematocoel, and morphology observed under confocal. (**G**) Haemocytes observed following Giemsa staining. (**H**) Three subpopulations of haemocytes morphologically identified and separated as agranulocytes (A), semi-granulocytes (SG), and granulocytes (G), by flow cytometry. (**I**) Representative immunofluorescence image shows the localization of *Cg*CHIP (green) in haemocytes and the nuclei stained with DAPI (blue). The localization region marked with yellow circles. (**J**) Bar graph shows the mean fluorescence intensity of *Cg*CHIP in the three haemocyte subpopulations. The per cell compartment was outlined, and the fluorescence intensity of positive signals within per cell was measured using ImageJ software. For each haemocyte subpopulation, the mean fluorescence value of ten cells from five fields were calculated as one replicate, and there were three replicates (*n* = 3). Abbreviations: Ara: agranulocytes; Semi-gra: semi-granulocytes; and Gra: granulocytes. (**K**) The percentages of three subpopulations in total haemocytes measured by flow cytometry (*n* = 3). (**L**) The bar graph shows the percentage of three haemocyte subpopulations (*n* = 3). *p*-values, * *p* < 0.05, ** *p* < 0.01, were calculated using a two-tailed, unpaired *t*-test.

**Figure 3 cells-13-01535-f003:**
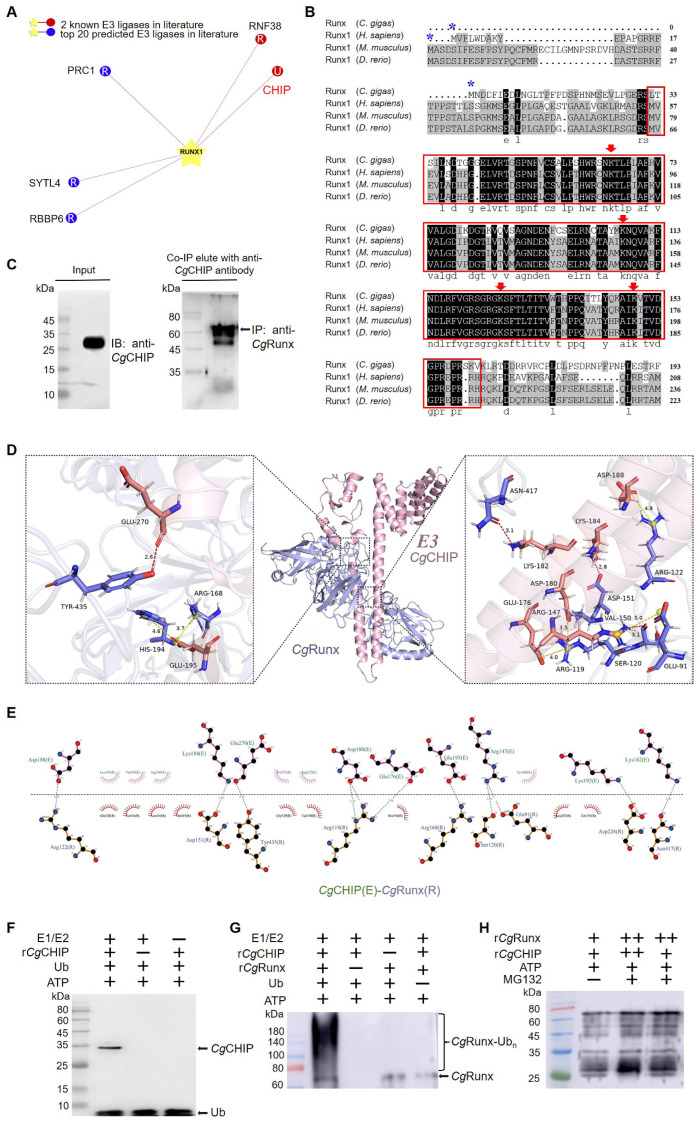
*Cg*CHIP targets the *Cg*Runx protein. (**A**) CHIP was known to interact with Runx1 as predicted by UbiBrowser 2.0 program. (**B**) The typical and conserved ubiquitination sites within the Runx protein. The Runt domain is marked with a red box. A Met-1 ubiquitination site is marked with a blue asterisk. Four conserved Lys ubiquitination sites are labeled with a red triangle. (**C**) Co-IP-based interaction detection of *Cg*CHIP and *Cg*Runx in oyster haemocytes. (**D**) Docking model analysis of *Cg*CHIP and *Cg*Runx. (**E**) The binding coefficients of *Cg*CHIP and *Cg*Runx protein interaction sites. (**F**) Ubiquitination activity of *Cg*CHIP detected with Western blotting in vitro. (**G**) *Cg*Runx ubiquitination assessed by Western blotting. (**H**) The levels of *Cg*Runx in oyster haemocytes treated with MG132 (20 μM), quantified by Western blotting.

**Figure 4 cells-13-01535-f004:**
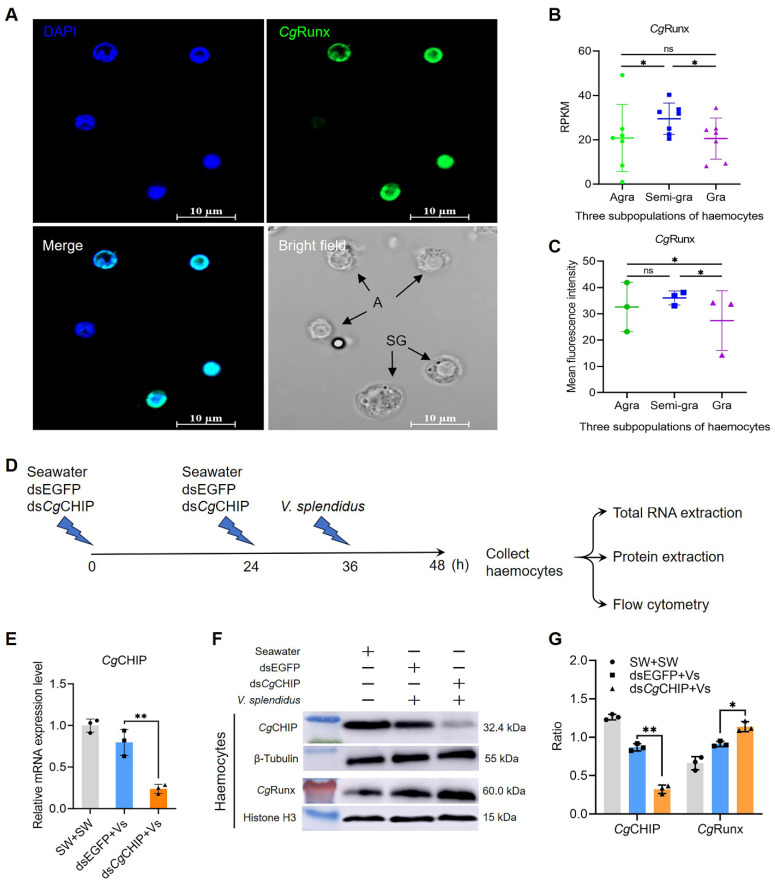
*Cg*CHIP enhances the ubiquitination and degradation of *Cg*Runx. (**A**) Representative immunofluorescence image shows the localization of *Cg*Runx (green) in haemocytes and the nuclei stained with DAPI (blue). (**B**) Transcriptome data analysis shows the mRNA transcripts of *Cg*Runx in the three haemocyte subpopulations (*n* = 7). *p*-values, * *p* < 0.05, were calculated using a one-way ANOVA with Dunnett’s correction for multiple comparisons. ns indicates no significant difference. (**C**) Bar graph shows the mean fluorescence intensity of *Cg*Runx in the three haemocyte subpopulations. (**D**) An injection cartoon of dsRNA in the interference assay. (**E**) The RNAi efficiency of *Cg*CHIP in haemocytes quantified via qRT-PCR, normalized to *Cg*EF1-α. Error bars show mean ± standard deviation (*n* = 3). *p*-values, ** *p* < 0.01, were calculated using a two-tailed, unpaired *t*-test. (**F**) Protein abundance of *Cg*CHIP (RNAi efficiency) and *Cg*Runx examined with Western blotting. (**G**) Gray analysis of protein band, normalized to β-Tubulin and Histone H3, respectively. *p*-values, * *p* < 0.05, ** *p* < 0.01, were calculated using a two-tailed, unpaired *t*-test.

**Figure 5 cells-13-01535-f005:**
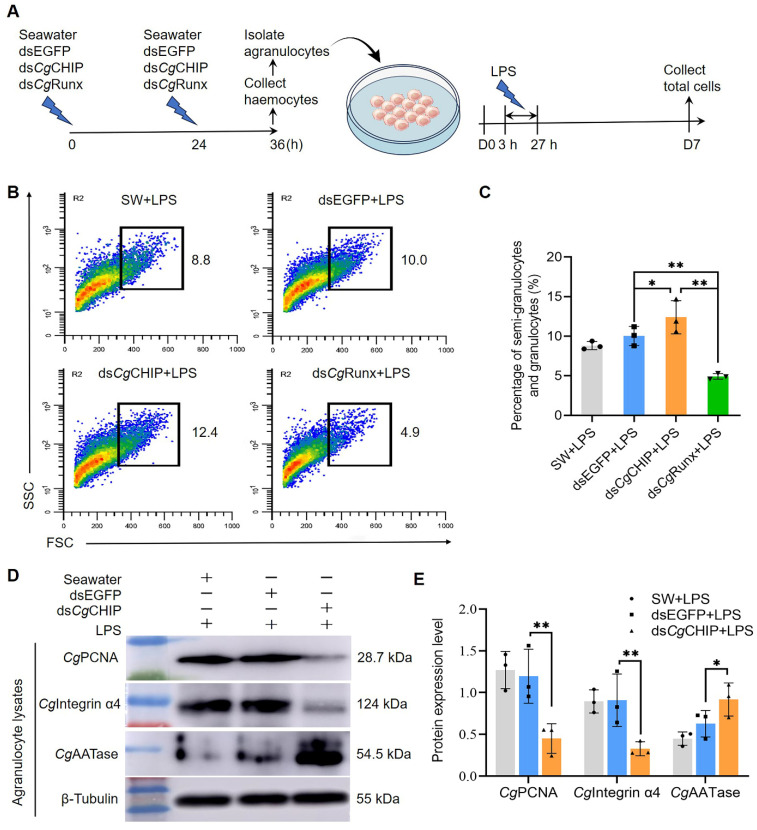
*Cg*CHIP inhibits agranulocyte differentiation. (**A**) Schematic of the induced differentiation in cultured agranulocytes. (**B**) Representative flow cytometry dot-plots show the gated semi-granulocyte and granulocyte populations differentiated from agranulocytes using the agranulocyte differentiation protocol. (*n* = 3). (**C**) The bar graph shows the percentage of differentiated agranulocytes (*n* = 3). * *p* < 0.05, ** *p* < 0.01, determined by a two-tailed Student’s *t*-test. (**D**,**E**) Protein expression levels of the proliferative marker *Cg*PCNA, immature agranulocyte marker *Cg*Integrin α4, and mature granulocyte marker *Cg*AATase, in agranulocytes. β-Tubulin was used as an internal control. Error bars show mean ± standard deviation (*n* = 3). * *p* < 0.05, ** *p* < 0.01, determined by a two-tailed Student’s *t*-test.

**Figure 6 cells-13-01535-f006:**
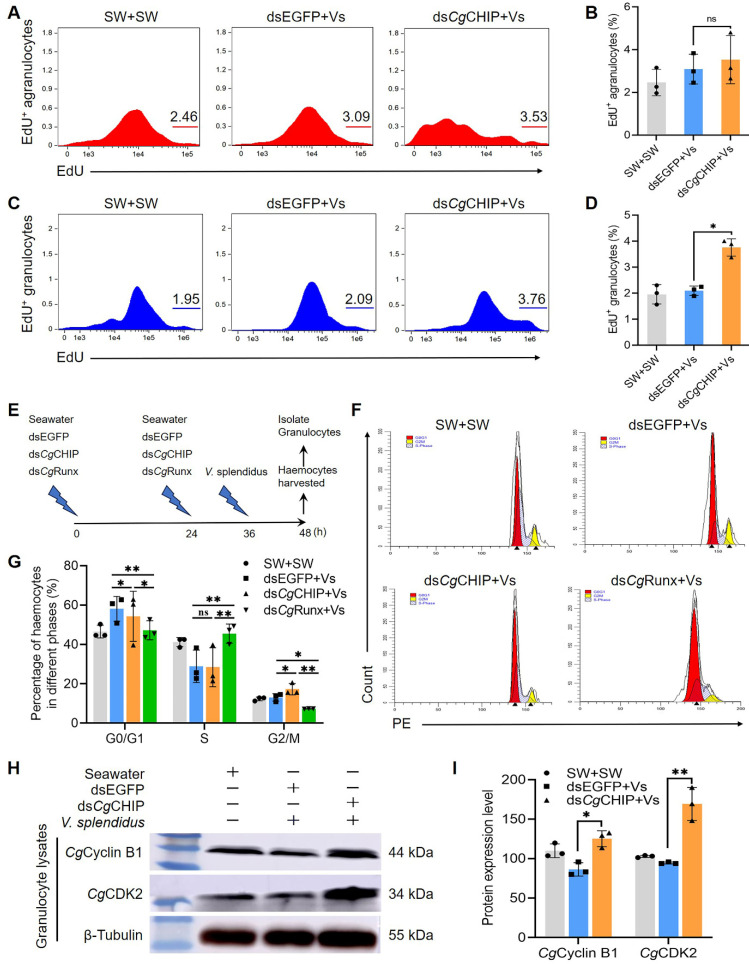
*Cg*CHIP inhibits granulocyte proliferation. (**A**) Representative flow cytometry peak diagrams show the proliferation rate of gated EdU labeling agranulocytes in total agranulocytes. (**B**) The bar graph shows the proliferation rate of agranulocytes (*n* = 3). (**C**) Representative flow cytometry peak diagrams showing the proliferation rate of gated EdU labeling granulocytes in total granulocytes. (**D**) The bar graph shows the proliferation rate of granulocytes (*n* = 3). * *p* < 0.05, determined by a two-tailed Student’s *t* test. (**E**) Schematic of granulocyte isolation for cell cycle and Western blotting analyses. (**F**) The percentage changes of granulocytes in different cell cycle phases. (**G**) The bar graph shows the percentage of agranulocytes in different cell cycle phases (*n* = 3). Error bars show mean ± standard deviation (*n* = 3). *p*-values, * *p* < 0.05, ** *p* < 0.01, were calculated using a one-way ANOVA with Dunnett’s correction for multiple comparisons. (**H**,**I**) Protein expression levels of proliferative genes *Cg*Cyclin B1 and *Cg*CDK2 in granulocytes. β-Tubulin was used as an internal control. Error bars show mean ± standard deviation (*n* = 3). The data shown are representative of three independent experiments. * *p* < 0.05, ** *p* < 0.01, determined by a two-tailed Student’s *t* test.

**Figure 7 cells-13-01535-f007:**
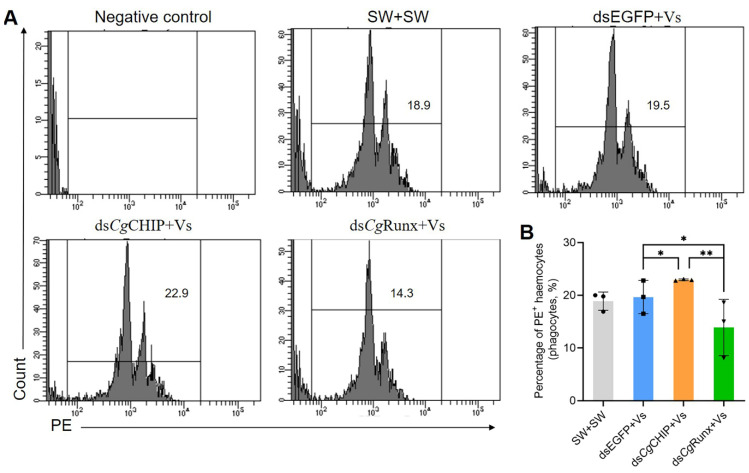
*Cg*CHIP attenuates phagocytosis in a *Cg*Runx-dependent manner. (**A**,**B**) Representative flow cytometry peak diagrams show the gated phagocytic haemocytes that are defined according to the red positive signal of latex beads. Phagocytic rate in haemocytes was defined by the percentage of phagocytic haemocytes taking in latex beads in total haemocytes. Error bars show mean ± standard deviation (*n* = 3). *p*-values were calculated using a one-way ANOVA with Dunnett’s correction for multiple comparisons. The asterisk * and ** indicated a significant difference at *p* < 0.05 and extremely significant difference at *p* < 0.01.

**Figure 8 cells-13-01535-f008:**
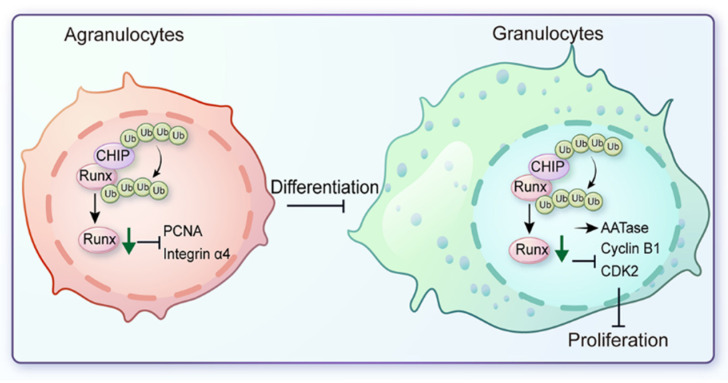
A graphical abstract. A conceptual framework for the ubiquitination and degradation of *Cg*Runx mediated by *Cg*CHIP, which inhibits the differentiation of agranulocytes and the proliferation of granulocytes in the Pacific oyster *C*. *gigas*.

## Data Availability

All study data are included in the article. For seeking other information and materials that are related to this project, please contact the corresponding author.

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
