# Peer review of "E3 Ubiquitin Ligase CHIP Inhibits Haemocyte Proliferation and Differentiation via the Ubiquitination of Runx in the Pacific Oyster"

_cells, 2024, doi:10.3390/cells13181535_

Round 1
Reviewer 1 Report
Comments and Suggestions for Authors
In their manuscript entitled “E3 ubiquitin ligase CHIP inhibits haemocyte proliferation and
differentiation via ubiquitination of Runx in the Pacific oyster” Dong and colleagues sought to demonstrate that CHIP ubiquitin ligase is an essential player in sea oyster immune response through haemocyte proliferation. The study identified the E3 ubiquitin ligase CHIP in oysters, suggesting it modulates haemocyte differentiation through Runx. Homologues of CgCHIP are already known to be implicated in immune cell proliferation and differentiation. Therefore, the novelty of the present work is reduced but remains of interest to scientists working with oysters.
Below are some comments for the authors to consider.
Major comments:
· Figure 2B: why does the mRNA level drop in SW control at 6 and 48hrs? Is “n=3” referring to technical or biological replicates? It would be ideal to have at least one positive and one negative control to those qPCRs (one gene that responds to infection and one that doesn’t).
· The expression of CgCHIP in the haemocytes fluctuates upon infection. How is the overall expression – or in other tissues – affected by the infection? Is the modulation of expression specific to haemocytes or a more general effect?
· Line 357: the authors claim that CgCHIP is evenly distributed in the cytoplasm and nucleus; however, from the selected picture in Figure 2I, it seems that CgCHIP staining is instead excluded from the nuclei. The authors may want to revise their statement or bring more evidence of nuclear expression (confocal microscopy (potentially with Z-stack) may help).
· The distribution pattern of CgCHIP in dots is interesting; could the authors comment further on this?
· Figure 2K-L: validation of the dsRNA should be displayed (qPCR for CgRunx and CgCHIP to demonstrate that the knockdown is efficient).
· Are the total number of haemocytes affected by the knockdown of CgRunx and CgCHIP?
· Figure 3C: negative control for IP is missing (IgG control IP) and blotting for both proteins (Runx and CHIP for both inputs and IP).
· Figure 3H: The difference between the last 2 lanes is not clear. To demonstrate that CgCHIP is indeed the ligase in vivo for CgRunx, the same experiment (or in vivo ubiquitination assay) should be performed in dsCHIP.
· Figure 4E: isn’t CgCHIP expression supposed to be increased upon infection (Figure 1)?
Minor comments:
· Line 296: is “star methods” a denomination used by MDPI?
· Figure 1A-B: maybe it would help the reader match the colours of the protein folds with the domains? What are the pink boxes?
· Lines 354 and 452: citations are not in the proper format.
· Figure 4E: “SW+SW” -> “dsEGFP+SW”?
· Figure 5B: the two top panels are identical.
English language (not limited to):
Line 122: “After washed with…” -> “being washed”?
Line 302: “it was comprised…” -> “it consists of…”?
Line 451: “proved” -> “proven”
Comments on the Quality of English LanguageThe manuscript would benefit from being carefully proof-read. Some examples of grammar mistakes below:
English language:
Line 122: “After washed with…” -> “being washed”?
Line 302: “it was comprised…” -> “it consists of…”?
Line 451: “proved” -> “proven”
Author Response
|
Response to Reviewer 1 Comments |
Thank you very much for taking the time to review this manuscript. Please find the detailed responses below and the corresponding revisions marked in red in the re-submitted files.
Comments of Reviewer 1:
In their manuscript entitled “E3 ubiquitin ligase CHIP inhibits haemocyte proliferation and differentiation via ubiquitination of Runx in the Pacific oyster” Dong and colleagues sought to demonstrate that CHIP ubiquitin ligase is an essential player in sea oyster immune response through haemocyte proliferation. The study identified the E3 ubiquitin ligase CHIP in oysters, suggesting it modulates haemocyte differentiation through Runx. Homologues of CgCHIP are already known to be implicated in immune cell proliferation and differentiation. Therefore, the novelty of the present work is reduced but remains of interest to scientists working with oysters.
- Figure 2B: why does the mRNA level drop in SW control at 6 and 48hrs? Is “n=3” referring to technical or biological replicates? It would be ideal to have at least one positive and one negative control to those qPCRs (one gene that responds to infection and one that doesn’t).
Response: Thanks for the reviewer’s comment and suggestion. The Pacific oysters are filter-feeding bivalves with an ‘open’ circulatory system in which the hemolymph bathes all the organs. The expression of genes in haemocytes shows minor fluctuations in response to the surroundings, so we set control groups for each time point [1].
“n=3” is referred biological replicates. The samples from three oysters were pooled together as one biological replicate, and there were three biological replicates for each tissue and haemocytes at each time point (n=3), which has been described in detail in lines 134-136.
We employ qRT-PCR technique to investigate the alteration of mRNA expression after immune stimulation, in which a reference gene (CgEF1-α) that does not respond to the stimulation was used for the negative control, and pattern recognition receptors (such as CgTLR2) and cytokines (such as CgIL17-1) that respond to the stimulation were regarded as positive controls, according to the previous studies [1, 2].
- The expression of CgCHIP in the haemocytes fluctuates upon infection. How is the overall expression – or in other tissues – affected by the infection? Is the modulation of expression specific to haemocytes or a more general effect?
Response: Thanks for the reviewer’s comment. CgCHIP was found to be significantly expressed in haemocytes, with a specific focus on regulating haemocyte fate determination, hence, we mainly investigate the alteration of CgCHIP in haemocytes after immune stimulation. Its response in other tissues post-infection was not analyzed in the present study.
- Line 357: the authors claim that CgCHIP is evenly distributed in the cytoplasm and nucleus; however, from the selected picture in Figure 2I, it seems that CgCHIP staining is instead excluded from the nuclei. The authors may want to revise their statement or bring more evidence of nuclear expression (confocal microscopy (potentially with Z-stack) may help).
Response 3: Thanks for the reviewer’s comment and suggestion. In the present study, the positive signals of CgCHIP labelled with Alexa Fluor 488 and the haemocyte nuclei labelled with DAPI were visible in green and blue, respectively, and they were co-localized (marked with a yellow circle in Figure 2I), which was added with symbols to display the necessary information.
- The distribution pattern of CgCHIP in dots is interesting; could the authors comment further on this?
Response 4: Studies have revealed that the aggregated protein dots known as aggresomes, consist of ubiquitin, proteasomes, E3 ligases, substrate proteins, and heat shock proteins, which play an important role in protein quality control and contribute to a swift response to stimuli and involved in signal transduction [3, 4]. CHIP targets multiple substrate proteins that function in cellular activities, such as cell cycle progression, the response to stress, signal transduction, and transcriptional regulation [4]. The dotted distribution of CgCHIP throughout the cytoplasm and nucleus in haemocytes may indicate its extensive roles in protein quality control and its rapid response to stimuli. These points have been added to the revised MS (lines 525-528).
- Figure 2K-L: validation of the dsRNA should be displayed (qPCR for CgRunx and CgCHIP to demonstrate that the knockdown is efficient).
Response 5: Thanks for the reviewer’s suggestion. The knockdown efficiency of CgCHIP in haemocytes was assessed by qRT-PCR and Western Blotting (Figures 4E-4G). The knockdown efficiency of CgRunx in haemocytes was assessed by Western Blotting, which has been added in Supplemental Figure 1.
- Are the total number of haemocytes affected by the knockdown of CgRunx and CgCHIP?
Response: Thanks for the reviewer’s comment. In the present study, we focused on the effects of CgRunx and CgCHIP on the proliferation and differentiation of haemocytes, and did not evaluate other cell fates, such as apoptosis. The change in the total number of haemocytes was not examined.
- Figure 3C: negative control for IP is missing (IgG control IP) and blotting for both proteins (Runx and CHIP for both inputs and IP).
Response: Thanks for the reviewer’s suggestion. The input of Runx and CHIP have been added in Supplemental Figure 2. We neglected to set up an IgG control in this study, but we are grateful for your reminder. Experimental design will be given full consideration in our future studies.
- Figure 3H: The difference between the last 2 lanes is not clear. To demonstrate that CgCHIP is indeed the ligase in vivo for CgRunx, the same experiment (or in vivo ubiquitination assay) should be performed in dsCHIP.
Response: Thanks for the reviewer’s comment and suggestion. In the second lane, there were increased cleavage bands at 25-35 kDa compared to the third lane, indicating an interaction between CgCHIP and CgRunx protein leading to a more robust degradation of CgRunx protein.
- Figure 4E: isn’t CgCHIP expression supposed to be increased upon infection (Figure 1)?
Response: Thanks for the reviewer’s comment. The mRNA level of CgCHIP in haemocytes fluctuates upon infection, and it slightly decreased at 12 h, which was also demonstrated in Figure 2B. These results may indicate that haemocytes tended to proliferate and differentiate during this phase of stimulation.
Minor comments:
- Line 296: is “star methods” a denomination used by MDPI?
Response: Thanks for pointing this out. We have corrected this issue in the revised manuscript.
- Figure 1A-B: maybe it would help the reader match the colours of the protein folds with the domains? What are the pink boxes?
Response: Thanks for the reviewer’s comment and suggestion. The pink box indicates a low complexity domain, which is an intrinsically disordered region that may be involved in protein flexibility, interactions, or specific biological processes. This information has been added in the result section 3.1 (line 310) and figure legend 1 section (lines 717-718) in the revised manuscript.
- Lines 354 and 452: citations are not in the proper format.
Response: Thanks for your careful checks and reminder. These have been corrected in the revised manuscript (lines 121, 404, 554).
- Figure 4E: “SW+SW” -> “dsEGFP+SW”?
Response: Thanks for the reviewer’s comment. The mRNA level in haemocytes decreased at 12 h after V. splendidus stimulation, as also demonstrated in Figure 2B.
- Figure 5B: the two top panels are identical.
Response: Thanks for your pointing out this mistake. We deeply apologize for our carelessness. It has been corrected in the revised Fig. 5A.
- English language (not limited to): Comments on the Quality of English Language. The manuscript would benefit from being carefully proof-read. Some examples of grammar mistakes below:
Line 122: “After washed with…” -> “being washed”?
Line 302: “it was comprised…” -> “it consists of…”?
Line 451: “proved” -> “proven”
Response: Thanks for your suggestion. These grammar mistakes have been revised in the re-submitted manuscript. Also, we have tried our best to polish the language in the revised manuscript. Thank you again for your correction.
- Fan, , Wang, W., Li, J., Cao, W., Li, Q., Wu, S., Wang, L., and Song, L. (2022). The truncated MyD88s negatively regulates TLR2 signal on expression of IL17-1 in oyster Crassostrea gigas. Dev Comp Immunol 133, 104446. https://doi: 10.1016/j.dci.2022.104446.
- Liu, Y., Wang, W., Sun, J., Li, Y., Wu, S., Li, Q., Dong, M., Wang, L., and Song, L. (2023). CgDM9CP-5-Integrin-MAPK Pathway Regulates the Production of CgIL-17s and Cgdefensins in the Pacific Oyster, Crassostrea gigas. J Immunol 210, 245-258. https://doi.org/10.4049/jimmunol.2200016.
- Pan, M., Zheng, Q., Wang, T., Liang, J., Zuo, C., Ding, R., Ai, H., Xie, Y., Si, D., Yu, Y., Liu, L., Zhao, M. (2021). Structural insights into Ubr1-mediated N-degron polyubiquitination, Nature 600, 334-338. https://doi: 10.1038/s41586-021-04097-8.
- Hatakeyama, S., and Nakayama, K.I. (2003). Ubiquitylation as a quality control system for intracellular proteins. J Biol Chem 134, 1-8. https://doi.org/10.1093/jb/mvg106.
Reviewer 2 Report
Comments and Suggestions for Authors
1. It will be important for the authors to show that mRNA mediated knockdown successfully reduced the levels of CgCHIP protein in cells. Although mRNA levels are reduced after 72 hours, it will be important to show more evidence which suggests that protein levels went down. For eg: western blots comparing protein levels before and after knockdown, fluorescent microscopy etc.
2. On a similar note, fig 2L showing the bar graph for the three sub-populations does not convincingly show that the there was a significant difference in the sub-populations
Author Response
|
Response to Reviewer 2 Comments |
Thank you very much for taking the time to review this manuscript. Please find the detailed responses below and the corresponding revisions in the re-submitted files.
Comments of Reviewer 2:
- It will be important for the authors to show that mRNA mediated knockdown successfully reduced the levels of CgCHIP protein in cells. Although mRNA levels are reduced after 72 hours, it will be important to show more evidence which suggests that protein levels went down. For eg: western blots comparing protein levels before and after knockdown, fluorescent microscopy etc.
Response: Thanks for the reviewer’s helpful comments and valuable suggestions. We agree with this comment. The knockdown efficiency of CgCHIP in haemocytes was confirmed at mRNA and protein levels by qRT-PCR and Western Blotting analysis (Figs. 4E-4G). Compared with the dsEGFP+Vs group, the protein levels of CgCHIP in the dsCgCHIP+Vs group went down, which was shown in the first bar in Fig. 4F.
- On a similar note, fig 2L showing the bar graph for the three sub-populations does not convincingly show that there was a significant difference in the sub-populations.
Response: Thanks for the reviewer’s comment. According to the reviewers' suggestions, we adjusted the presentation method for a clearer display of the result. Please see the re-submitted Fig. 2L.
Round 2
Reviewer 2 Report
Comments and Suggestions for Authors
Recommended changes were made. Changes are satisfactory.